# Oxepinamide F biosynthesis involves enzymatic D-aminoacyl epimerization, 3H-oxepin formation, and hydroxylation induced double bond migration

Liujuan Zheng[1,3], Haowen Wang[1,3], Aili Fan [2] & Shu-Ming Li [1✉]

Oxepinamides are derivatives of anthranilyl-containing tripeptides and share an oxepin ring and a fused pyrimidinone moiety. To the best of our knowledge, no studies have been reported on the elucidation of an oxepinamide biosynthetic pathway and conversion of a quinazolinone to a pyrimidinone-fused 1H-oxepin framework by a cytochrome P450 enzyme in fungal natural product biosynthesis. Here we report the isolation of oxepinamide F from *Aspergillus ustus* and identification of its biosynthetic pathway by gene deletion, heterologous expression, feeding experiments, and enzyme assays. The nonribosomal peptide synthase (NRPS) OpaA assembles the quinazolinone core with D-Phe incorporation. The cytochrome P450 enzyme OpaB catalyzes alone the oxepin ring formation. The flavoenzyme OpaC installs subsequently one hydroxyl group at the oxepin ring, accompanied by double bond migration. The epimerase OpaE changes the D-Phe residue back to L-form, which is essential for the final methylation by OpaF.

[1] Institut für Pharmazeutische Biologie und Biotechnologie, Fachbereich Pharmazie, Philipps-Universität Marburg, Robert-Koch Straße 4, 35037 Marburg, Germany. [2] College of Life Science and Technology, Beijing University of Chemical Technology, North Third Ring Road 15, Chaoyang District, 100029 Beijing, People's Republic of China. [3] These authors contributed equally: Liujuan Zheng, Haowen Wang. ✉email: shuming.li@staff.uni-marburg.de

Oxepinamides are a class of oxepins with a fused pyrimidinone ring and were mainly found in fungi. For example, oxepinamides F, G, and H were isolated from *Aspergillus puniceus*[1,2], varioloid A and varioxepine A *from Paecilomyces variotii*[3,4], dihydrocinereain from *Aspergillus carneus*[5], circumdatin A/B from *Aspergillus ochraceus*[6], and versicoloid A/B from *Aspergillus versicolor*[7] (Fig. 1a). Some oxepinamides show high affinity to liver X receptors (LXRs) and are potential agents for the treatment of Alzheimer's disease, atherosclerosis, diabetes, and inflammation[1,2]. Oxepinamides are usually derived from quinazolinones with an anthranilyl (Ant) residue in common. They differ from each other by incorporation of two other varying amino acids and additional modifications. Until now, four trimodular nonribosomal peptide synthases (NRPSs) for assembling quinazolinones containing Trp, Ala, and Gly have been functionally characterized (Fig. 1b). In comparison to AldpA[8] and CtqA[8], TqaA[9] and FmqA[10] have an additional epimerization (E) domain in the Trp module, which is responsible for the conversion of L- to D- tryptophan in fumiquinazoline (FQF). As shown in Fig. 1a, phenylalanine, valine, leucine, and isoleucine residues are often found in the oxepinamide structures. To the best of our knowledge, genes responsible for quinazolinones with these amino acids have not been reported prior to this study.

In comparison to 1*H*-oxepins with three C=C in the oxepin ring, varioxepine A, varioloid A, oxepinamide F and G feature a 3*H*-oxepin structure with two C=C in the ring and one *exo* C=N bond (Fig. 1a). To the best of our knowledge, despite the intriguing structural features and biological activities, studies on the biosynthesis of oxepinamides, especially the formation of the 3*H*-oxepin ring, have not been reported yet.

We identify in this study an oxepinamide (*opa*) biosynthetic gene cluster (BGC) in *Aspergillus ustus* by bioinformatic analysis. Gene deletion, heterologous expression, feeding experiments, and in vitro assays with purified enzymes prove the biosynthetic steps and the 3*H*-oxepin formation by consecutive ring expansion and regio- and stereospecific hydroxylation. Furthermore, the D-phenylalanyl epimerase OpaE converts the D-Phe residue back to L-form for the last methylation step to form oxepinamide F.

## Results

**Identification of the *opa* BGC.** Oxepinamide F (**1**) was isolated, together with its nonmethylated congener oxepinamide E (**2**) from a rice culture of *A. ustus* 3.3904. Their structures were confirmed by mass spectrometry (MS), nuclear magnetic resonance (NMR), optical rotation, and circular dichroism (CD) analyses (NMR data are listed in Supplementary Tables 5–8 and spectra in Supplementary Figs 7–30. CD spectra are given in Supplementary Fig. 3)[1]. Typical NMR signals for 3*H*-oxepins were observed at $\delta_H$ 6.7 (d, 7.3 Hz), $\delta_H$ 5.5 (t, 7.1 Hz), $\delta_H$ 6.2 (dd, 10.0, 7.0 Hz), and $\delta_H$ 5.8 ppm (d, 10.1 Hz), as well as $\delta_C$ 144, 104, 130, and 128 ppm in their spectra.

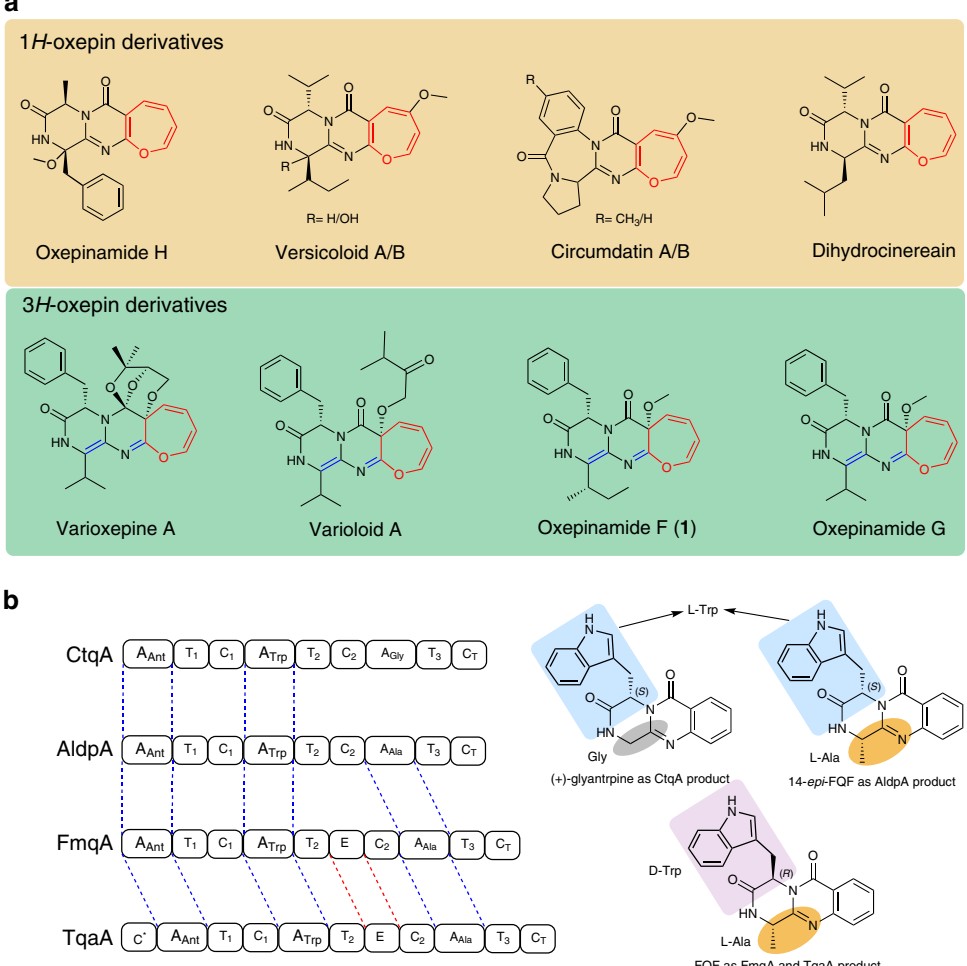

**Fig. 1 Oxepinamides and backbone enzymes. a** Examples of fungal oxepinamides including 1*H*-oxepins and 3*H*-oxepins. **b** Known NRPSs for quinazolinone assembling.

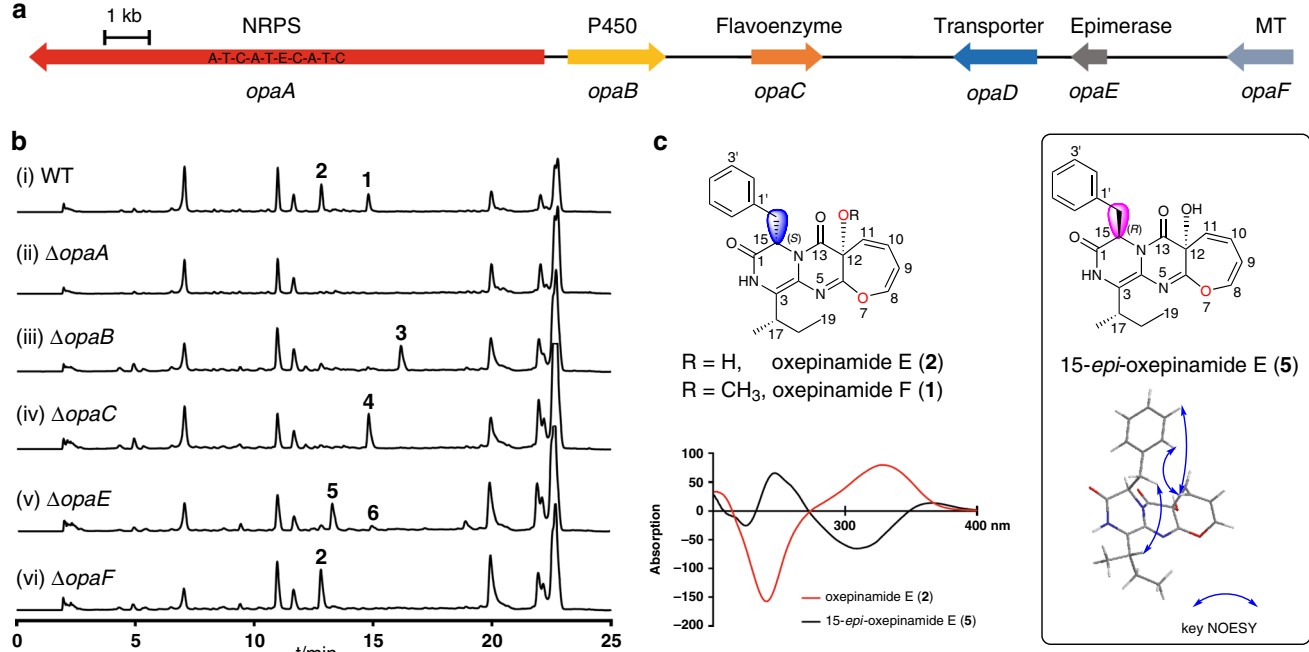

**Fig. 2 Biosynthetic genes of oxepinamide F and intermediates. a** The oxepinamide biosynthetic gene cluster (*opa*) from *A. ustus* 3.3904. **b** HPLC analysis at 254 nm of the extracts after 7 day cultivation. **c** Structures of oxepinamide F (**1**), E (**2**), and 15-*epi*-oxepinamide E (**5**), as well as the CD spectra of **2** and **5** (200–400 nm). Blue double-headed arrows represent NOESY interactions.

To find the biosynthetic genes of **1**, we sequenced the *A. ustus* 3.3904 genome, carried out prediction for putative BGCs by using AntiSMASH[11], and compared them with the published data[12]. The core structure of **1** is proposed to be assembled from Ant, Phe, and Ile by a NRPS containing at least three A-T-C (A: adenylation, T: thiolation, C: condensation) modules[13]. **1** differs from **2** merely in the methyl group at OH-12, implying that an *O*-methyltransferase (*O*-MeT) should be involved in its biosynthesis. Comprehensive sequence analysis revealed the presence of at least 11 genes for putative NRPSs. Two of them, KIA75458 and KIA75688, contain three A-T-C modules. Furthermore, a gene coding for a putative *O*-MeT (KIA75453) was only found neighboring to the gene for KIA75458. Thus, the KIA75458-related BGC was the best candidate for the **1** and **2** biosynthesis.

Inspection of the genomic neighborhood of these candidate genes in *A. ustus* revealed the presence of a putative BGC containing six genes *opaA*–*opaF*, coding for the putative proteins KIA75458–KIA75453 in the database (Fig. 2a). Sequence analysis and comparison suggested their functions as putative cytochrome P450 enzyme (OpaB, KIA75457), FAD-dependent oxygenase (OpaC, KIA75456), transporter (OpaD, KIA75455), epimerase (OpaE, KIA75454), and *O*-MeT (OpaF, KIA75453) (Supplementary Table 1).

Further sequence analysis revealed that KIA75458, named OpaA in this study, has a domain architecture of A-T-C-A-T-E-C-A-T-C (Fig. 2a). To prove its function, *opaA* in *A. ustus* was replaced with a hygromycin B resistance cassette by using a split marker gene replacement protocol[14]. The generated mutants were verified by PCR (Supplementary Fig. 1) and cultivated in rice media. High-performance liquid chromatography (HPLC) analysis of the culture extract of the Δ*opaA* mutant revealed the abolishment of both **1** and **2** production (Fig. 2b), proving its involvement in their biosynthesis. Deletion of *opaF* abolished **1**, but not **2** production, indicating that OpaF acts as a methyltransferase for the conversion of **2** to **1** (Fig. 2b). Based on above results, the *opa* gene cluster is indeed responsible for biosynthesis of **1** and **2**.

The oxepin ring in **1** and **2** was proposed to be formed by oxidative benzene ring expansion[1,15]. However, it is unknown whether the P450 enzyme OpaB, the oxidase OpaC or both are responsible for this conversion. To figure out their functions, we firstly deleted *opaB* from *A. ustus* genome. Deletion of *opaB* led to the abolishment of **1** and **2**, together with the accumulation of a new peak **3** at 16.2 min with a $[M + H]^+$ ion at *m/z* 362.189 (Fig. 2b). Typical signals for oxepins were not observed in the NMR spectra of **3**. Extensive interpretation of the spectroscopic data including NMR and CD spectra and comparison with known compounds[16] proved **3** to be protuboxepin K, a quinazolinone derivative of Ant, D-Phe, and L-Ile (Fig. 3)[17].

Bioinformatic analysis and comparison with known proteins revealed that OpaA consists of a deduced $A_{Ant}$-$T_1$-$C_1$-$A_{Phe}$-$T_2$-E-$C_2$-$A_{Ile}$-$T_3$-$C_T$ domain structures, similar to CtqA, AldpA, FmqA, and TqaA. OpaA has the same $A_{Ant}$ domain for anthranilic acid activation (Fig. 1b). As aforementioned, the E domain in FmqA and TqaA is responsible for incorporation of a D-Trp residue in FQF. In analogy, the presence of an E domain in the second module of OpaA and the D-Phe residue in **3** imply that the second A domain is for activating of L-Phe, which is then converted to D-form by the E domain. The third A domain is responsible for L-Ile activation and the terminal $C_T$ domain for cyclization and releasing the NRPS product. Thus, OpaA is a quinazolinone synthase using different amino acids from those listed in Fig. 1b, which expands clearly quinazolinone structure diversity.

**OpaB functions as an oxepinase.** The results from deletion experiments provide unambiguous evidence that OpaB is responsible for the expansion of the benzene to the oxepin ring. For further understanding of its function, *opaB* was amplified from genomic DNA and cloned into pYH-*gpdA*-*afpyrG*[18] via homologous recombination in yeast[19] for heterologous expression in *Aspergillus nidulans*. The obtained plasmid pLZ61 was linearized by SwaI and integrated into the genome of *A. nidulans* LO8030[20] (Supplementary Fig. 2). Feeding **3** in the obtained overexpression

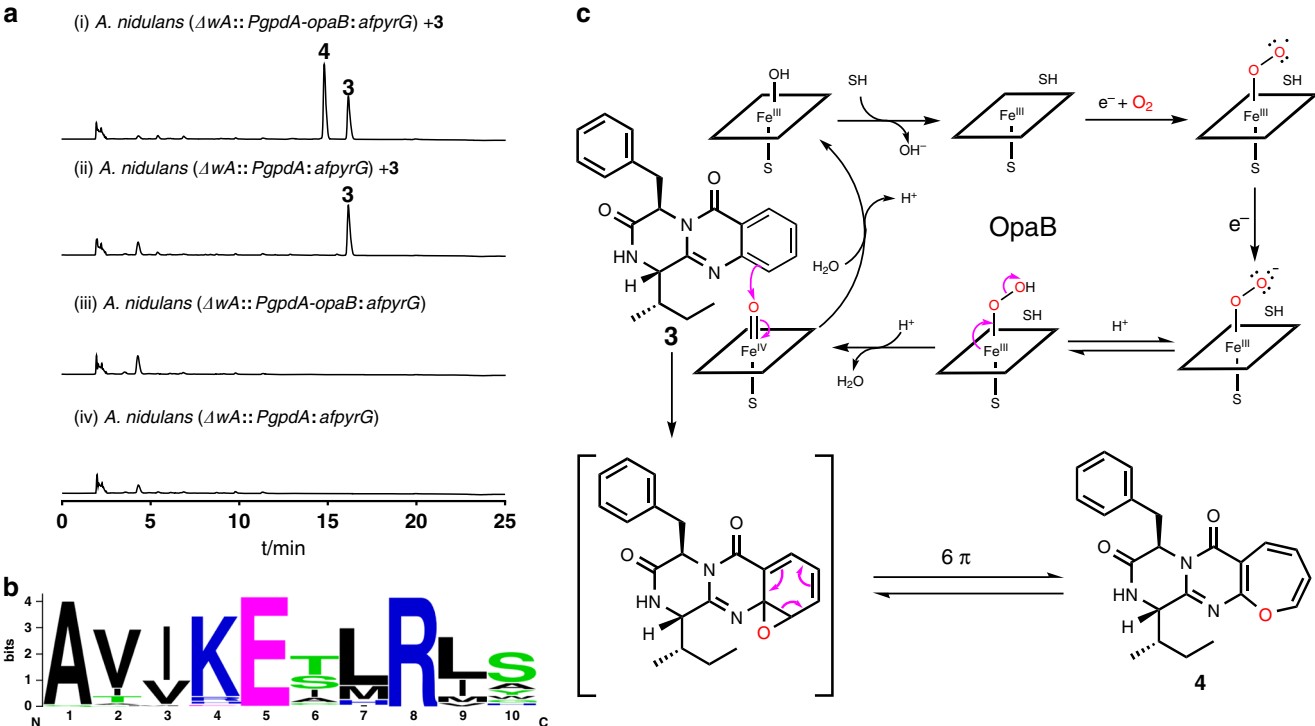

**Fig. 3 Functional verification of the oxepinase OpaB. a** HPLC analysis of the conversion of **3** to **4** in the *opaB* expression strain of *A. nidulans* at 254 nm. **b** A sequence logo for the conserved EXXR motif in OpaB using sequences of 96 P450 enzymes (Supplementary Table 9). **c** Proposed mechanism for oxepin formation catalyzed by OpaB (SH: substrate).

strain *A. nidulans* LZ61 led to the detection of compound **4**, which was not observed in a negative control. **4** was confirmed to be protuboxepin A by NMR analysis and by comparison of its optical rotation value with that reported in literature[21]. This confirmed that OpaB alone is responsible for the oxepin ring formation (Fig. 3a). Sequence alignment indicated that the conserved P450 motif ExxR is present as ETMR in OpaB (Fig. 3b and Supplementary Table 9)[22–24]. In analogy to other P450 catalyzed alkane hydroxylations and alkene epoxidations[25,26], we postulated the electrophilic oxoferryliron (Fe$^{IV}$=O) as the active oxygen intermediate in the OpaB reaction. Attacking of the oxoferryliron species by the nucleophilic benzene double bond in **3** would result in the formation of an arene oxide, which is in rapid spontaneous equilibrium with the oxepin **4** through 6π electrocyclic ring opening (Fig. 3c)[15,27]. The oxepin form is expected to be more stable at room temperature than its arene oxide[27,28]. Only the 1*H*-oxepin **4** was detected as OpaB poduct in this study, which differs from an oxepin intermediate important for both phenylacetate degradation and tropone biosynthesis. In those cases, the isomerase PaaG forms a stable 3*H*-oxepin from a labile 1*H*-oxepin[29]. On the other hand, both 1*H*- and 3*H*-oxepin derivatives listed in Fig. 1 were isolated as stable fungal metabolites. Phylogenetic analysis of OpaB with 51 known P450 enzymes from bacteria and fungi indicates clearly the presence of different clades (Supplementary Fig. 4). The bacterial P450 enzymes catalyzing diverse reactions, including the two hydroxylases P450cin and P450cam with trace oxepin formation activity[28], build a distinct clade from fungal enzymes. The majority of the fungal P450s in the phylogenetic tree catalyzes hydroxylations of diverse substrates. OpaB is located near to the epoxidase AtaY, but far away from the oxepinase AtaF, both involved in the biosynthesis of acetylaranotin in *Aspergillus terreus*[30].

**OpaC catalyzes hydroxylation accompanied by double bond migration.** Comparing the planar structures, **2** differs from **4** in

the OH-12 and two different double bonds. Sequence analysis suggested OpaC to be a FAD-containing monooxygenase and could be a good candidate for a consecutive hydroxylation at C12 of the oxepin ring. The double bonds could be shifted during the hydroxylation reaction. To verify its function biochemically, *opaC* was amplified from complementary DNA (cDNA) and cloned into pET28a (+) for overexpression in *Escherichia coli*. The recombinant *N*-terminally His$_6$-tagged protein was purified to near homogeneity with a yield of 3.5 mg per liter culture (Fig. 4b). To our surprise, one product peak **5** at 13.3 min, instead of **2** at 12.8 min, was detected in the incubation mixture of **4** with the purified OpaC in the presence of NADPH (Fig. 4a). **5** shares the same UV visible light absorption and mass spectral features with **2**. NMR data and CD data supported that **5** and **2** are diastereomers and differ from each other merely in the configuration at C15. **5** was, therefore, named 15-*epi*-oxepinamide E (Fig. 2c). Deletion of *opaC* from the *A. ustus* genome led indeed to the accumulation of **5** (Fig. 2b). Biochemical characterization revealed that OpaC also used NADH as a cofactor, but with a significantly reduced activity (Fig. 4a). No substrate consumption was observed in the incubation mixture of **3** and OpaC (Supplementary Fig. 5), proving the prerequisite of the oxepin ring for an acceptance by OpaC. The kinetic data of the OpaC reaction with **4** in the presence of NADPH fit well to a typical velocity equation with substrate inhibition[31,32]. An apparent $K_M$ value at $0.43 \pm 0.04$ mM, a turnover number ($k_{cat}$) at $0.16 \pm 0.01$ s$^{-1}$, the catalytic efficiency ($k_{cat}/K_m$) at $0.37$ mM$^{-1}$ s$^{-1}$ and a substrate inhibition constant ($K_i$) at $0.39 \pm 0.03$ mM (Fig. 4c) were calculated by using the software GraphPad Prism 6.0.

Based on these results, a mechanism with a C4a-hydroperoxyflavin intermediate[33] was postulated for the OpaC reaction (Fig. 4d). The oxidized flavin Fl$_{ox}$ is converted to its reduced form Fl$_{red}$ by external electron donor NADPH. Subsequent reaction of Fl$_{red}$ with O$_2$ produces the electrophilic reagent C4a-hydroperoxyflavin. The elimination of the proton at

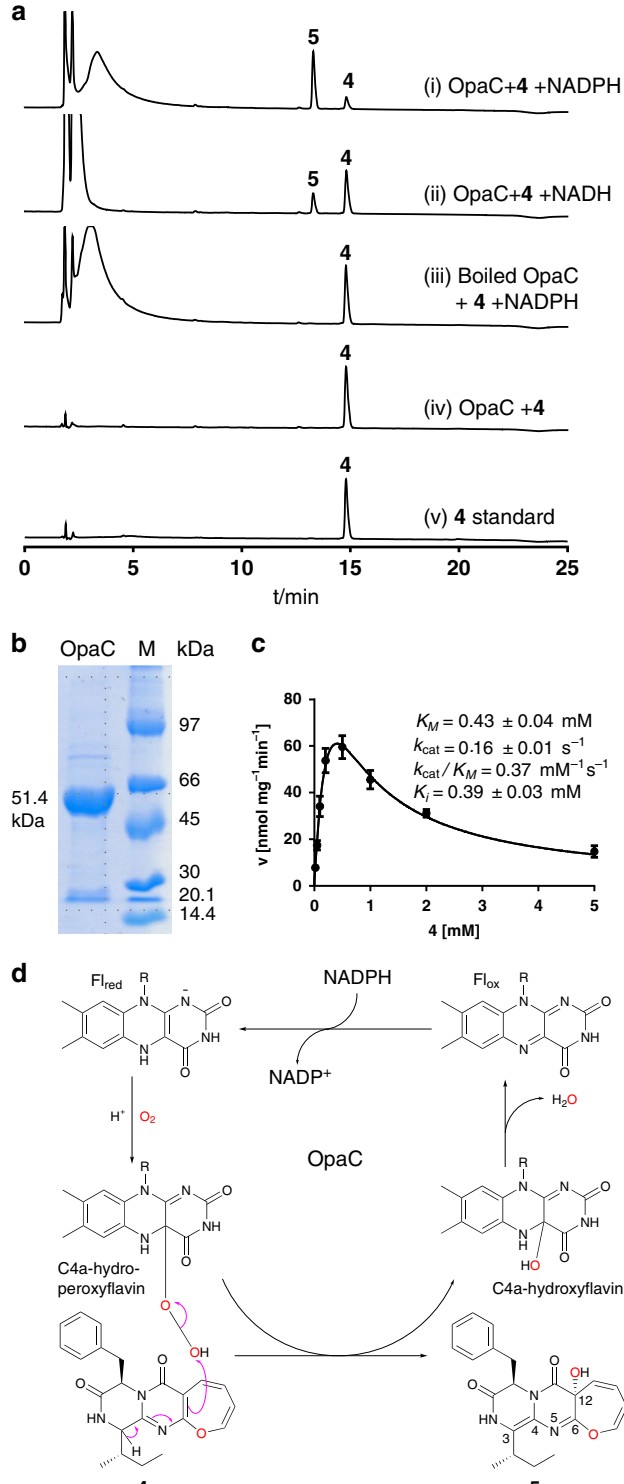

**Fig. 4 Proof of the OpaC function. a** HPLC analysis of in vitro assays of **4** with OpaC at 254 nm. **b** SDS-PAGE analysis of the purified OpaC. The experiments were repeated twice. **c** Determination of the kinetic parameters of the OpaC reaction. The error bars represent standard errors of velocity and the standard errors of mean (SEMs) are given as ±values ($n = $ six independent experiments). **d** Proposed mechanism for the OpaC reaction with a C4a-peroxyflavin as a key intermediate.

C3 in **4** results in the double bond migration and attack on the C4a-hydroperoxyflavin, leading to the formation of **5** and C4a-hydroxyflavin. The latter one is then regenerated to $Fl_{ox}$ by removal of one water molecule for the next reaction cycle.

Sequence analysis and biochemical investigation revealed that OpaC belongs to the well-studied class A flavin-dependent monooxygenases[34]. Phylogenetic analysis of representatives from this group (Supplementary Fig. 6) indicates the presence of at least three clades. OpaC is located closely together with AspB in the biosynthesis of asperlicins and PhqK in that of paraherquamides. AspB catalyzes the hydroxylation at C3 of an indole moiety, resulting in the formation of a hexahydropyrrolo[2, 3-b] indole framework[35]. PhqK converts an indole ring to a 2-keto indoline ring via a postulated arene oxide intermediate[36]. In both cases, the C4a-hydroperoxyflavin species was proposed to serve as oxygen transferring agent, consistent with other class A flavin monooxygenases and our mechanistic proposal.

**D-Phenylalanyl epimerization catalyzed by OpaE**. As aforementioned, **2** differs from **5** only in the configuration at C15. The $S$-configuration at this position in **2** corresponds to that of L-Phe, which was epimerized to $R$-configuration by the E domain of OpaA in **3**. Conversion of **5** to **2** would need an epimerase like OpaE. Deletion of *opaE* led indeed to the abolishment of **1** and **2** production and main accumulation of **5** with a trace amount of a methylated product **6** (15-*epi*-oxepinamide F, Fig. 2b). To prove OpaE function in vitro and to understand the epimerization mechanism, the coding region of *opaE* was cloned and overexpressed as described for OpaC. HPLC analysis of the reaction mixture of **5** with the recombinant OpaE revealed **2** as the mere product peak (Fig. 5), proving unequivocally its function as an epimerase. Configuration change at C15 is a prerequisite for further methylation to the final product **1**, because only trace amounts of **5** were converted to its methylated product **6** in $\Delta opaE$ mutant (Fig. 2b). Determination of the kinetic parameters of the OpaE reaction with **5** via Michaelis–Menten equation gave a $K_M$ value of $1.41 \pm 0.05$ mM, a turnover number ($k_{cat}$) of $0.28 \pm 0.01$ s$^{-1}$ and the catalytic efficiency ($k_{cat}/K_M$) at $0.20$ mM$^{-1}$ s$^{-1}$ (Fig. 5c).

Incubation of **5** and OpaE in Tris-HCl buffer containing $D_2O/H_2O$ (9:1) and subsequent analysis on liquid chromatography–mass spectrometry (LC–MS) led to detection of the shifted $[M + H]^+$ isotopic pattern of **2** ($[M + H]^+$ 395.184), which is 1 Da larger than that in $H_2O$ ($[M + H]^+$ 394.177) (Fig. 5d). Incubation of **2** in Tris-HCl buffer containing $D_2O/H_2O$ (9:1) did not change the isotopic pattern. These results proved the involvement of an enol intermediate in the OpaE-catalyzed epimerization, as proposed in Fig. 5e[37].

## Discussion

Taken together, we identified the *opa* cluster for the oxepinamide F biosynthesis. OpaA was proven to activate Ant, L-Ile, and L-Phe, change the configuration of L-Phe to D-Phe by an epimerase domain, and assemble the quinazolinone **3**. The benzene ring in the Ant residue of **3** was expanded to an oxepin ring in **4** by the P450 enzyme OpaB alone. The regio- and stereospecific hydroxylation of **4** catalyzed by OpaC was accompanied by double bond migration from C4-N5 and C6-C12 in **4** to C3-C4 and N5-C6 in **5**, leading to the conversion of *1H*-oxepin to *3H*-oxepin system. The $R$-configuration in **5** was changed to $S$-configuration by the single epimerase OpaE for the final methylation of the OH-12 by the $O$-methyltransferase OpaF, to the end product **1** (Fig. 6).

Oxepin rings also play important roles in the bacterial degradation of phenylacetic acid and the biosynthesis of tropone

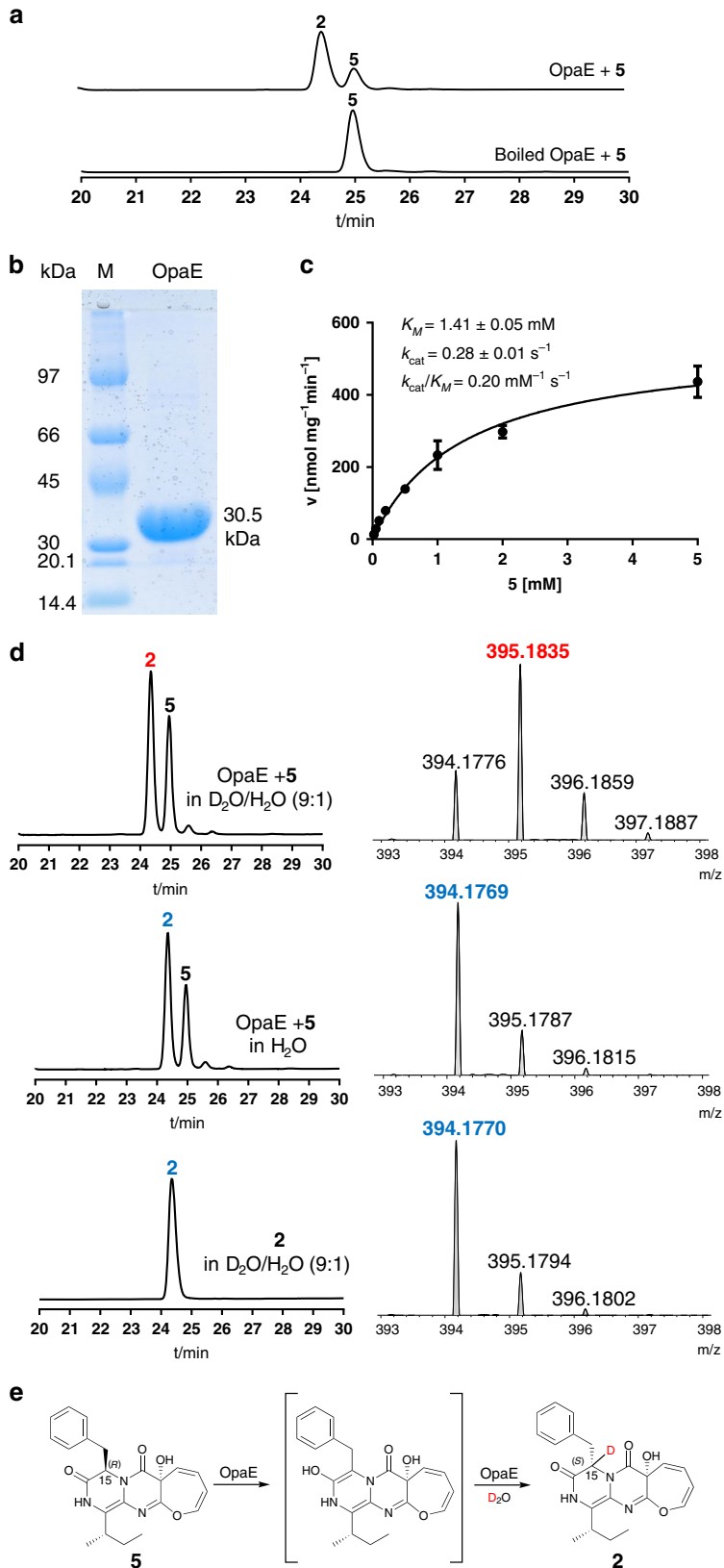

**Fig. 5 In vitro verification of OpaE as a d-phenylalanyl epimerase. a** HPLC analysis of in vitro assays with OpaE at 254 nm. **b** SDS-PAGE analysis of the purified OpaE. The experiments were repeated twice. **c** Determination of the kinetic parameters for OpaE with **5**. The error bars represent standard errors of velocity and SEMs are given as ± values ($n$ = six independent experiments). **d** LC–MS monitoring of the OpaE reactions in $D_2O/H_2O$ and negative controls. **e** Proposed mechanism for the OpaE reaction.

**Fig. 6 Proposed biosynthetic pathway for oxpinamide F in *Aspergillus ustus*.** Ant, L-Phe and L-Ile are assembled to protuboxepin K (**3**) by the NRPS OpaA with epimerization at the phenylalanyl residue. The oxepinase OpaB catalyzes the oxepin ring formation in protuboxepin A (**4**). The regio- and stereospecific hydroxylation of **4** by OpaC is accompanied by a double bond migration, leading to the conversion of a 1*H*-oxepin to a 3*H*-oxepin system in **5**. The D-Phe configuration in **5** is changed to L-Phe in **2** by the single epimerase OpaE. The *O*-methyltransferase OpaF catalyzes the conversion of **2** to the end product **1** by methylation of the hydroxyl group at C-12.

natural products that both depend on the multicomponent epoxidase PaaABCE and oxepin isomerase PaaG[29,38]. Stok et al.[28] observed the formation of a simple oxepin derivative as a minor side product in the hydroxylation of *tert*-butyl benzene by two bacterial P450 enzymes, which usually use the monoterpenes camphor and 1, 8-cineole as natural substrates, respectively. Wang and coworkers reported the formation of a dihydrooxepin ring in the biosynthesis of an epipolythiodioxopiperazine by three enzymes, the epoxidase AtaF, the acetyltransferase AtaH, and the cyctochrome P450 enzyme AtaY[30]. The oxepin ring was formed from an acetoxylated cyclohexan-diene structure.

In peptide biosynthesis, conversion of L- to D-form amino acids during the peptide biosynthesis is usually catalyzed by epimerase domains of NRPSs or by radical *S*-adenosylmethionine-dependent enzymes in microorganisms[39,40]. No single enzyme was reported to catalyze the conversion of D- to L-form of an amino acid residue. To the best of our knowledge, an oxepinamide biosynthetic gene cluster that includes the characterization of involved biosynthetic enzymes and reactions steps has not been reported before. Furthermore, conversion of an amino acid residue from L- to D-configuration and then back to the L-configuration by two different epimerases for structural modification has not been described in literature.

## Methods

**Genome sequencing and sequence analysis**. The genome of *A. ustus* 3.3904 was sequenced by Genewiz (Suzhou, China) using Nova-seq6000/X-ten (Illumina). Initial prediction and analysis of biosynthetic gene clusters were carried out by using AntiSMASH[41]. Prediction of the enzyme function was performed with the online BLAST programmer (http://blast.ncbi.nlm.nih.gov). The genomic DNA sequence of the *opa* cluster (Supplementary Table 1) reported in this study corresponds well to that depicted at GenBank under accession number: JOMC01000153.1.

The phylogenetic trees were created by MEGA version 7.0 (http://www.megasoftware.net). Protein sequence alignments were performed with the program ClustalW (https://www.genome.jp/tools-bin/clustalw) to identify strictly conserved amino acid residues.

**Strains, media, and growth conditions**. *Escherichia coli* DH5α and BL21(DE3) cells were grown in LB medium (1% NaCl, 1% tryptone, and 0.5% yeast extract) at 37 °C. In all, 50 μg mL$^{-1}$ ampicillin or 50 μg mL$^{-1}$ kanamycin were supplemented for cultivation of recombinant strains.

*Saccharomyces cerevisiae* HOD114-2B cells were grown in YPD medium (1% yeast extract, 2% peptone and 2% glucose, 1.5% agarose was used for plate). The SC-uracil medium (6.7 g L$^{-1}$ yeast nitrogen base with ammonium sulfate, 650 mg L$^{-1}$ CSM-His-Leu-Ura (MP Biomedicals), 20 mg L$^{-1}$ His and 60 mg L$^{-1}$ Leu, pH 6.2 − 6.3, 1.5% agarose was used for plate) with 2.0% glucose was used for selection.

Fungal strains used in this study are listed in Supplementary Table 2. *Aspergillus ustus* 3.3904 was purchased from China General Microbiological Culture Collection Center (Beijing, China) and cultivated in rice medium (20 g Alnatura long-grain rice with 30 mL H$_2$O in 250 mL flask) at 25 °C for of secondary metabolite production.

*Aspergillus nidulans* LO8030 and derivatives were grown at 37 °C on GMM medium (1.0% glucose, 50 mL L$^{-1}$ salt solution, 1 mL L$^{-1}$ trace element solution, and 1.6% agar) for sporulation and transformation with appropriate nutrition as required. The salt solution comprises (w/v) NaNO$_3$, 1.04% KCl, 1.04% MgSO$_4$ · 7H$_2$O, and 3.04% KH$_2$PO$_4$. The trace element solution contains (w/v) 2.2% ZnSO$_4$ · 7H$_2$O, 1.1% H$_3$BO$_3$, 0.5% MnCl$_2$ · 4H$_2$O, 0.16% FeSO$_4$ · 7H$_2$O, 0.16% CoCl$_2$ · 5H$_2$O, 0.16% CuSO$_4$ · 5H$_2$O, 0.11% (NH$_4$)$_6$Mo$_7$O$_{24}$ · 4H$_2$O, and 5% Na$_4$EDTA.

**Genomic DNA isolation**. The *A. ustus* 3.3904 and *A. nidulans* mycelia were dried on filter paper and transferred into 2 mL Eppendorf tubes. After addition of 400 μL LETS buffer (10 mM Tris-HCl pH 8.0, 20 mM EDTA pH 8.0, 0.5% SDS, and 0.1 M LiCl) and four glass beads (2.85 mm in diameter), the tubes were vigorously mixed for 4 min. After addition of another 300 μL LETS buffer, the mixtures were treated with 700 μL phenol: chloroform: isoamyl alcohol (25: 24: 1). The genomic DNA in the aqueous phase was precipitated by addition of 900 μL absolute EtOH, followed by centrifugation at 17,000 x *g* for 30 min and washing with 70% EtOH. The obtained DNA as pellet was dried at 55 °C for 15 min and dissolved in 50–100 μL distilled H$_2$O.

**RNA isolation from *A. ustus* 3.3904 and cDNA synthesis**. For this purpose, *A. ustus* 3.3904 was cultivated in rice medium at 25 °C for 7 days. The mycelia were collected by washing with 50 mL H$_2$O and subsequent centrifugation. Fungal RNA Mini kit (VWR OMEGA bio-tek E.Z.N.A) was used for RNA extraction following the standard protocol provided by the manufacturer. For cDNA synthesis, the ProtoScript II First Strand cDNA Synthesis kit (BioLabs) was used with Oligo-dT as primers.

The deduced polypeptide from the coding region of *opaC* obtained from cDNA comprises 457 amino acids, lacking the five residues from 248 to 252 in KIA75456:

**PCR amplification, gene cloning, and plasmid construction**. Primers and plasmids used in this study are listed in Supplementary Tables 3 and 4, respectively. Seqlab GmbH (Göttingen, Germany) synthesized the PCR primers. Phusion® High-Fidelity DNA polymerase from New England Biolabs (NEB) were used for PCR amplification, which was carried out on a T100TM Thermal cycler from Bio-Rad by following the manufacturer´s suggestion for temperature profiles.

**Genetic manipulation in *A. ustus* 3.3904 and cultivation of deletion mutants**. To get germlings for protoplast preparation, fresh spores of *A. ustus* 3.3904 were inoculated in a 250 mL flask containing 50 mL LMM medium and incubated at 230 rpm and 30 °C. After 11 h, the germlings were harvested by centrifugation at 2280 x *g* in 4 °C for 15 min and washed with distilled $H_2O$. The pellets were suspended in 10 mL osmotic buffer (1.2 M $MgSO_4$ in 10 mM sodium phosphate, pH 5.8) containing 40 mg lysing enzyme from *Trichoderma harzianum* (Sigma) and 30 mg yatalase from *Corynebacterium sp.* OZ-21 (OZEKI Co., Ltd.) and incubated at 100 rpm and 30 °C for 10 h. The formation of protoplasts was monitored by controlling under microscope. The mixture was then transferred into a 50 mL falcon tube and overlaid gently with 10 mL of trapping buffer (0.6 M sorbitol in 0.1 M Tris-HCl, pH 7.0). The protoplasts appeared as an interface between the two buffer systems after centrifugation at 2280 x *g* and 4 °C for 15 min. This interface was transferred carefully into a 15 mL falcon tube. The protoplasts in this phase were collected by centrifugation at 2280 x g, and resuspended in 100 μL of STC buffer (1.2 M sorbitol, 10 mM $CaCl_2$ in 10 mM Tris-HCl, pH 7.5).

For transformation, the obtained protoplasts were incubated with DNA samples (2–3 μg in 8–10 μL) on ice for 50 min. 1.25 mL of PEG solution (60% PEG 4000, 50 mM $CaCl_2$, 50 mM Tris-HCl, pH 7.5) was then added, gently mixed and incubated at room temperature for 30 min. In all, 5 mL STC buffer were added into the mixture and spread on plates with SMM bottom medium (1.0% glucose, 50 mL L$^{-1}$ salt solution, 1 mL L$^{-1}$ trace element solution, 1.2 M sorbitol and 1.6% agar) containing 100 μg mL$^{-1}$hygromycin B. The plates were overlaid softly by SMM top medium (1.0% glucose, 50 mL L$^{-1}$ salt solution, 1 mL L$^{-1}$ trace element solution, 1.2 M sorbitol, and 0.8% agar) containing 50 μg mL$^{-1}$hygromycin B. After cultivating at 30 °C for 3–4 days, the fungal colonies were transferred on fresh PDA plates (PD medium with 3% agar) containing 100 μg mL$^{-1}$ hygromycin B for selection. The potential transformants were verified via PCR amplification (Supplementary Fig. 1), cultivated in rice medium at 25 °C for 7 days and extracted with EtOAc. After evaporation, the extracts were dissolved in dimethyl sulfoxide (DMSO) and subjected to LC–MS for analysis.

**Heterologous expression in *A. nidulans***. In this study, we used *A. nidulans* LO8030[42] as expression host. The protoplast preparation procedure was similar to that for *A. ustus* 3.3904 as mentioned above. The germlings were shaken with 40 mg lysing enzyme from *Trichoderma harzianum* (Sigma) and 20 mg yatalase from *Corynebacterium sp.* OZ-21 (OZEKI Co., Ltd.) at 100 rpm and 37 °C for 3 h. pLZ61 containing the cytochrome P450 gene *opaB* was transformed into *A. nidulans* LO8030 to create the expression strain LZ61. The transformants were verified by PCR (Supplementary Fig. 2) and cultivated for biotransformation with protuboxepin K (**3**).

**Overproduction and purification of OpaC and OpaE**. The *opaC* and *opaE* expression plasmids pLZ62 and pLZ63 were transferred separately into *E. coli* BL21 (DE3). Terrific Broth (TB) medium (2.4% yeast extract, 2.0% tryptone, 0.4% glycerol, 0.1 M phosphate buffer, pH 7.4) was used for cultivation. OpaC overproduction was induced with 0.5 mM IPTG at 16 °C for 16 h, and OpaE with 1 mM IPTG at 20 °C for 20 h. The recombinant His$_6$-tagged proteins were purified by Ni-NTA affinity chromatography (Qiagen, Hilden). The protein concentration was determined on a Nanodrop 2000c spectrophotometer (Thermo Scientific, Braunschweig, Germany) and analyzed by sodium dodecyl sulfate-polyacrylamide gel electrophoresis (SDS-PAGE) (Figs. 4b and 5b). Protein yields of 3.5 and 20 mg per liter bacterial culture were calculated for OpaC and OpaE, respectively.

**In vitro assays of OpaC and OpaE**. To test the enzyme activity of OpaC, the reaction mixtures (50 μL) containing 50 mM Tris-HCl (pH 7.5), 5 mM NADPH, 1 mM protuboxepin A (**4**), 10 μg (3.9 μM) of the purified recombinant OpaC were incubated at 30 °C for 30 min. The reactions were terminated by addition of 50 μL methanol.

For OpaE, the reaction mixtures (50 μL) containing 50 mM Tris-HCl (pH 7.5), 1 mM 15-*epi*-oxepinamide E (**5**), 5 μg (3.2 μM) of the purified recombinant OpaE were incubated at 37 °C for 30 min. The reactions were terminated by addition of 50 μL methanol.

For deuterium labeling experiment with OpaE, the stock solutions of the assay components in $H_2O$ were diluted with $D_2O$ to a final $D_2O/H_2O$ ratio of 9:1 in 50 mM Tris-HCl. The reaction mixtures were incubated and analyzed on LC–MS as described for standard assays.

**Determination of kinetic parameters**. Enzyme assays for determination of the kinetic parameters for OpaC (50 μL) contained 50 mM Tris-HCl (pH 7.5), 5 mM NADPH, 4 μg (1.6 μM) of the purified recombinant OpaC and protuboxepin A (**4**) at final concentrations from 0.025 to 2.5 mM. The incubations were carried out at 30 °C for 30 min. After extraction of the reaction mixtures with EtOAC and evaporation of the solvent to dryness, the residues were dissolved in DMSO and analyzed on HPLC.

For OpaE, 50 μL reaction mixtures contained 50 mM Tris-HCl (pH 7.5), 2 μg (1.3 μM) of the purified recombinant OpaE and compound 15-*epi*-oxepinamide E (**5**) at final concentrations from 0.02 to 2 mM. The incubations were carried out at 37 °C for 30 min. After addition of MeOH and centrifugation, the supernatants were analyzed on HPLC. The data presented in Figs. 4c and 5c were obtained from six independent experiments. SEMs are given as ± values (*n* = six independent experiments).

Kinetic parameters were determined by nonlinear regression using the software GraphPad Prism 6.0 via substrate inhibition velocity equation for OpaC and Michaelis–Menten equation for OpaE.

**Large-scale fermentation, extraction, and isolation of secondary metabolites**. The strains were cultivated in 0.75 kg rice (1.75 L) for isolation. The rice culture was extracted with equal volume of EtOAc for three times. The EtOAc extracts were concentrated under reduced pressure to afford the crude extracts for further purification.

To isolate oxepinamide F (**1**) and E (**2**), the crude extract of *A. ustus* wildtype was subjected to silica gel (230-400 mesh) column chromatography, by using a gradient of $CH_2Cl_2/CH_3OH$ (100:0-0:100) to give eight fractions (Fr.1-Fr.8). Fr.4 was purified on a semi-preparative HPLC (ACN/$H_2O$), leading to 38.42 mg of oxepinamide F (**1**), 57.24 mg of oxepinamide E (**2**). Similarly, 77.68 mg protuboxepin K (**3**) was obtained from a culture of the Δ*OpaB* mutant, 79.24 mg protuboxepin A (**4**) from a culture of the Δ*OpaC* mutant, 23.88 mg 15-*epi*-oxepinamide E (**5**) 16.88 mg 15-*epi*-oxepinamide F (**6**) from a culture of the Δ*OpaE* mutant and 15.78 mg oxepinamide E (**2**) from a culture of the Δ*OpaF* mutant.

**Feeding experiments in the *opaB* expression strain *A. nidulans* LZ61**. To figure out the function of the cytochrome P450 enzyme OpaB, *opaB* was overexpressed in *A. nidulans* with the *gpdA* promoter from *A. fumigatus*[18]. Ten milliliter PDB with 75 μL 0.5 mg mL$^{-1}$ riboflavin and 10 μL 0.5 mg mL$^{-1}$pyridoxine in a 25 mL erlenmeyer flask were inoculated at 230 rpm and 30 °C. One milligram of protuboxepin K (**3**) (8 mg mL$^{-1}$DMSO) was added into 3 days old culture of the transformant *A. nidulans* LZ61. 16 h later, 1 mL culture was extracted with 1 mL EtOAc. The extracts were dried and dissolved in 100 μL DMSO for LC–MS analysis. Strain LZ61 without feeding protuboxepin K (**3**) and *A. nidulans* assembled empty vector with or without feeding protuboxepin K (**3**) were used as negative controls.

**HPLC analysis and metabolite isolation**. Fungal extracts were analyzed on an Agilent HPLC series 1200 (Agilent Technologies) by using an Eclipse XDB-C18 column (Agilent Technologies, 5 μm, 4.6 × 150 mm) and ACN/$H_2O$ as elution solvents. A linear gradient from 10 to 90% ACN in $H_2O$ containing 0.1% (v/v) HCOOH in 20 min was used. After washing with 100% ACN for 5 min, the column was equilibrated with 10% ACN for another 5 min. A photodiode array detector was used for detection and the absorptions at 254 nm are illustrated in this study.

For product isolation, a Multospher 120 RP-18 column (5 μm, 10 × 250 mm) was used on the same HPLC system, with the same elution solvents at a flow rate of 2 mL min$^{-1}$. Separation was performed by isocratic elution with 45–70% ACN in $H_2O$ containing 0.1% (v/v) HCOOH for 10–20 min, and.

**LC–MS and MS analysis**. Extracts were also analyzed on an Agilent HPLC 1260 series system equipped with a Bruker microTOF QIII mass spectrometer by using a Multospher 120 RP-18-5μ column (5 μm, 250 × 2 mm). Separation was accomplished in a 40 min linear gradient from 5 to 100% ACN in $H_2O$, both containing 0.1% (v/v) HCOOH at a flow rate of 0.25 mL min$^{-1}$. The column was then washed with 100% ACN for 5 min followed by equilibration with 5 % ACN for 10 min. Positive ions were scanned in the range of *m/z* 100–1500 under the following conditions: capillary voltage with 4.5 kV, collision energy with 8.0 eV and electrospray ionization. Mass calibration was achieved by using sodium formate in each run. Data collection and analysis were carried out with the Compass DataAnalysis 4.2 software (Bruker Daltonik, Bremen, Germany).

**NMR analysis**. Samples in high purity were dissolved in DMSO-*d*$_6$ or CDCl$_3$ and subjected to JEOL ECA-500 or ECA-400 (JEOL, Akishima, Tokyo, Japan) for taking NMR spectra. MestReNov.9.0.0 (Mestrelab Research, Santiago de Compostella, Spain) was used for spectral processing.

**CD spectroscopic analysis**. The samples were dissolved in $CH_3OH$ and measured in the range of 200–400 nm by using a 1 mm path length quartz cuvette (Hellma Analytics, Müllheim, Germany) on a J-815 CD spectrometer (Jasco Deutschland GmbH, Pfungstadt, Germany). The CD spectra are shown in Supplementary Fig. 3.

**Measurement of optical rotations**. Jasco DIP-370 at 25 °C equipped with the D-line of the sodium lamp at $\lambda = 589.3$ nm was utilized to measure the optical rotations of the isolated compounds in $CHCl_3$ or $CH_3OH$. The polarimeter was calibrated with the respective solvent before the measurement.

**Reporting summary**. Further information on research design is available in the Nature Research Reporting Summary linked to this article.

## Data availability

The information of *opa* gene cluster can be obtained from JOMC01000153.1 in the NCBI database. The authors declare that all relevant data supporting the finding of this study are available within the paper and its Supplementary Information files. All data are available from the corresponding author on reasonable request. Source data are provided with this paper.

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

## Acknowledgements

We thank Rixa Kraut, Lena Ludwig-Radtke, and Stefan Newel for taking MS and NMR spectra, Lena Ludwig-Radtke and Yiling Yang for technical assistance. This project was funded in part by the Deutsche Forschungsgemeinschaft (DFG, INST 160/620-1). Liu-juan Zheng (201604910536) is a scholarship recipient from the China Scholarship Council. The CD spectrometer was provided by the core facility for protein biochemistry and spectroscopy in the Institute of Cytobiology.

## Author contributions

S-M.L. directed the research. L.Z. conducted genetic manipulation and biochemical studies. H.W. performed compound isolation and structure elucidation. L.Z., H.W., A.F., and S-M.L. designed the experiments, analyzed the data, and wrote the manuscript. L.Z. and H.W. contributed equally to this work.

## Funding

## Competing interests

The authors declare no competing interests.
