## [Peer Review File · Nature Communications]

REVIEWER COMMENTS

Reviewer #1 (Remarks to the Author):

The manuscript by Zheng et al. describes the investigation of fungal oxepinamide F biosynthesis. This work includes identification and heterologous expression of the corresponding gene cluster as well as feeding experiments, structural characterization of key intermediates and enzyme assays to further support the proposed biosynthetic pathway. The manuscript is clearly structured and very well written. Overall, the data quality appears to be high and all content is nicely presented, including necessary controls to support the conclusions of the paper. The very thorough analyses and unambiguous data presented in this work therefore provides detailed insights into the NRPS assembly line biosynthesis of the oxepinamide core peptide and further tailoring reactions by redox enzymes, a methyltransferase and an epimerase. The comprehensive work will undoubtedly be of high interest to a broader scientific community, e.g., researchers interested in natural product biosynthesis, medicinal chemistry or enzymology and I thus support publication of this work in Nature Communications.

However, before publication, the flavin monooxygenase mechanism and the corresponding description should be corrected. In addition, the authors should more thoroughly discuss the oxepin-epoxide tautomerization and some of their results. Finally, various minor issues should be addressed, as pointed out below.

Major points:

- The flavin catalysis in Fig. 4 and the corresponding discussion contain a few mistakes. Only one molecule of NADPH (+ H⁺) is required for the two-electron reduction of the flavin (shown are two NADPH). Also the C4a-peroxyflavin should be depicted in the protonated form, i.e. remove the brackets from the distal oxygen. Finally, remove the exiting proton in the step that yields C4a-hydroperoxyflavin (both educt and product have a neutral charge). Maybe also consider to remove the coloring of the H-atoms and only label the oxygens. In that regard, the description of the mechanism on p. 5 should be fixed: "Subsequent reaction of Fl_{red} with O₂ produces C4a-hydroperoxyflavin, which can easily lose one proton to form C4a-peroxyflavin. The peroxide anion serves then as a nucleophilic reagent and is attacked by the electron pair of the double bond between C6 and C12 of 4 to produce C4a-hydroxyflavin. 4 was converted to an intermediate with a hydroxyl group at C12, which is then transformed to 5 by double bond migration". First, the C4a-hydroperoxyflavin acts as an electrophilic agent that is attacked by the substrate; this is correctly shown in the figure but incorrectly described in the text. The reaction is formally an electrophilic substitution reaction. Deprotonation to the nucleophilic C4a-peroxyanion occurs for flavin monooxygenases that catalyze Baeyer-Villiger oxygenations, which is not the case here. Second, I would suggest to propose a concerted mechanism, i.e. the elimination of the proton at C3 results in the double bond migration and attack on the C4a-hydroperoxyflavin (again, this is correctly shown in the figure but described differently in the text).
- If the length requirements of the journal permits it, I suggest to add a more thorough discussion of the results. This includes for example a brief discussion on flavoenzyme phylogeny. To my understanding, OpaC belongs to the well-studied class A flavin monooxygenases that typically feature mobile flavins, see e.g., PMID: 24361254 or PMID: 19944667. One of these references (or a similar one) should be cited here. This would also provide additional support for the proposed mechanism, as so far all class A monooxygenases exclusively utilize C4a-(hydro)peroxide species for catalysis (other flavin monooxygenases may instead use N5-oxygen adducts for catalysis, see PMID: 32066967 and PMID: 24162851). I would also consider to add a phylogenetic tree to the SI that includes other characterized members of class A monooxygenases. A similar approach could be used for the P450 epoxidase OpaB. What are the closest characterized homologues of this enzyme?
- The oxepin-epoxide tautomerization should be discussed better. Typically, arene oxides are in rapid spontaneous equilibrium with their oxepins. Which compound is favored depends on the

solvent and conjugants mainly in alpha position to my understanding (see DOI: 10.1002/anie.196703851, this reference should also be cited). The authors should discuss if it is known to what extent the different oxepinamides are in equilibrium with their epoxide tautomers. Compound stability might be an important clue here, as epoxides are more reactive and unstable than the oxepins. I would expect the 3H-oxepinamides such as oxepinamide F to be more stable, as the exo double bond formation should completely prevent tautomerization to the epoxide in contrast to the 1H-oxepinamides such as oxepinamide H. This might be a similar strategy as employed by PaaG of phenylacetate degradation and tropone natural product biosynthesis that also forms a stable 3H-oxepin moiety from a labile 1H-oxepin (PMID: 31689071).

Minor points:

- Spelling of oxepine in general should be oxepin? Are both valid?
- Abstract: correct "oxapinamide".
- p. 2 first paragraph: correct to "potential agents for the treatment of..., and inflammation".
- p. 2 first paragraph: Please write fumiquinazoline in full before using the abbreviation FQF.
- p. 2 second paragraph: correct to "structural features".
- p. 3, Results: write "A. ustus" in italics.
- Fig. 1: the compound number for oxepinamide F could be added.
- Fig. 2: Please indicate the wavelength of the shown chromatograms.
- Fig. 3: The conversion of the epoxide to the oxepin should be fully reversible under physiological conditions, which should be correctly depicted here (i.e. an enzyme should normally not be needed). Or is there any clear reason/evidence for why this isomerization is irreversible? Please correct and discuss, see also comments and suggested references above.
- p. 3, first line: Correct to "...we sequenced the A. ustus..."
- p. 3, second last paragraph: "m/z 362.1888". I suggest to delete the last digit and round the number to 362.189, as typically such a high mass accuracy is not achieved (also apply to other m/z values).
- p. 4, last sentence: This description of the catalytic mechanism of OpaB does not fit the mechanism shown in Fig. 3c. To my understanding, epoxidation could either be catalyzed by an electrophilic hydroxyperoxy-ferric iron (FeIII-OOH) or by an electrophilic oxoferryl-iron (FeIV=O), which should be pointed out here and also in the figure legend (see for example PMID: 9520404 or PMID: 15365901).
- Fig. 4: add wavelength to HPLC chromatograms. Also, the enzyme does not appear to follow typical Michaelis-Menten kinetics, there seems to be some sort of inhibitory effect with substrate conditions above 0.5 mM. This should at least be pointed out. Please also use the term "apparent KM" here.
- p. 5, first paragraph: correct to monooxygenase. Also, what does "planer" mean? Please change to "...5 shares the same UV visible light absorption and mass spectral features with 2".
- p. 5 second paragraph: correct to "epimerization".
- p. 6: correct to "phenylacetic acid". Also, this statement is somewhat unclear, I suggest changing to "Oxepin rings also play important roles in the bacterial degradation of phenylacetic acid and the biosynthesis of tropone natural products that both depend on the multicomponent epoxidase PaaABCE and oxepin isomerase PaaG" (consider adding PMID: 31689071 in addition to ref 29).
- p. 6: The following statement should be removed or revised "Thus, OpaB represents the first single P450 to catalyze the direct conversion of a benzene to an oxepine ring by one step conversion." Basically every enzyme that epoxidizes an aromatic ring also generates an oxepin ring that is spontaneously formed from the epoxide, see also comments above.
- p. 6: correct to "...isomerase from microorganisms involved..."
- Fig. 5: add wavelength to HPLC chromatograms.
- Fig. 6 is not correctly depicted so I cannot assess the content. Please fix.
- p. 7: "the cultures were extracted with EtOAc, dissolved in DMSO and subjected to HPLC ..." I suspect the authors mean that EtOAc extracts were rotavaped and the extracted compounds then redissolved in DMSO before analysis? Please explain more clearly.
- p. 7: Remove space character between 15-epi-oxepinamide.
- p. 8: correct to "...the CD spectra are shown..."

- SI Table 7: The icons for COSY, NOESY and HMBC couplings are not properly displayed, please fix.

Reviewer #2 (Remarks to the Author):

The manuscript describes the identification of the oxepinamide F biosynthetic gene cluster and characterization of several of the steps in the pathway. The authors used a combination of in vivo and in vitro data to functionally assign the enzymes. For the former, gene inactivation often led to the isolation of biosynthetic intermediates that were thoroughly characterized by spectroscopy. The latter involved the functional assignment of flavin-dependent oxygenase and an unexpected epimerase.

Overall, the scientific rigor is excellent.

One of the key assignments is the P450 protein that catalyzes the insertion of the oxygen atom to make the oxepine that is characteristic of the family. This result along with the other data make this a very interesting paper with high significance and will have a broad impact in the field. The manuscript is, in general, well written. Once the following minor concerns are addressed, the paper is ready for publication.

One of the figures was cut off apparently due to formatting? There are other formatting errors that need to be corrected.

The method used for the deuterium wash-in experiment should be included in the materials and methods section.

Reviewer #3 (Remarks to the Author):

This is a very nice and well-written manuscript describing the biosynthesis of a class of natural products from *Aspergillus*. However, it is my opinion that this work is better suited for a more specialised journal such as *Antimicrobial Agents and Chemotherapy*. The approaches used are well-known, and the biosynthesis of similar compounds have been elucidated. There are no unusual enzymatic reactions reported here. Thus while this work is rigorous and well done, I do not believe that it is of interest to the very general readership of *Nature Communications*.

Reviewer(s)' Comments to Author:

Reviewer #1

The manuscript by Zheng et al. describes the investigation of fungal oxepinamide F biosynthesis. This work includes identification and heterologous expression of the corresponding gene cluster as well as feeding experiments, structural characterization of key intermediates and enzyme assays to further support the proposed biosynthetic pathway. The manuscript is clearly structured and very well written. Overall, the data quality appears to be high and all content is nicely presented, including necessary controls to support the conclusions of the paper. The very thorough analyses and unambiguous data presented in this work therefore provides detailed insights into the NRPS assembly line biosynthesis of the oxepinamide core peptide and further tailoring reactions by redox enzymes, a methyltransferase and an epimerase. The comprehensive work will undoubtedly be of high interest to a broader scientific community, e.g., researchers interested in natural product biosynthesis, medicinal chemistry or enzymology and I thus support publication of this work in Nature Communications.

Many thanks for the positive comments.

However, before publication, the flavin monooxygenase mechanism and the corresponding description should be corrected. In addition, the authors should more thoroughly discuss the oxepin-epoxide tautomerization and some of their results. Finally, various minor issues should be addressed, as pointed out below.

1. Major points:

1.1 The flavin catalysis in Fig. 4 and the corresponding discussion contain a few mistakes. Only one molecule of NADPH (+ H⁺) is required for the two-electron reduction of the flavin (shown are two NADPH). Also the C4a-peroxyflavin should be depicted in the protonated form, i.e. remove the brackets from the distal oxygen. Finally, remove the exiting proton in the step that yields C4a-hydroperoxyflavin (both educt and product have a neutral charge). Maybe also consider to remove the coloring of the H-atoms and only label the oxygens. In that regard, the description of the mechanism on p. 5 should be fixed: "Subsequent reaction of Fl_{red} with O₂ produces C4a-hydroperoxyflavin, which can easily lose one proton to form C4a-peroxyflavin. The peroxide anion serves then as a nucleophilic reagent and is attacked by the electron pair of the double bond between C6 and C12 of **4** to produce C4a hydroxyflavin. **4** was converted to an intermediate with a hydroxyl group at C12, which is then transformed to **5** by double bond migration". First, the C4a-hydroperoxyflavin acts as an electrophilic agent that is attacked by the substrate; this is correctly shown in the figure but incorrectly described in the text. The reaction is formally an electrophilic substitution reaction. Deprotonation to the nucleophilic C4a-peroxyanion occurs for flavin monooxygenases that catalyze Baeyer-Villiger oxygenations, which is not the case here. Second, I would suggest to propose a concerted mechanism, i.e. the elimination of the proton at C3 results in the double bond migration and attack on the C4a-hydroperoxyflavin (again, this is correctly shown in the figure but described differently in the text).

Response 1.1 Thanks a lot for the insightful comments in the mechanism of flavin monooxygenases. We have corrected all the mistakes in Figure 4 and rephrased the description on the mechanism as following.

The oxidized flavin Fl_{ox} is converted to its reduced form Fl_{red} by external electron donor NADPH. Subsequent reaction of Fl_{red} with O₂ produces the electrophilic reagent C4a-hydroperoxyflavin. The elimination of the proton at C3 in **4** results in the double bond migration and attack on the C4a-hydroperoxyflavin, leading to the formation of **5** and C4a-hydroxyflavin. The latter one is then regenerated to Fl_{ox} by removal of one water molecule for the next reaction cycle.

1.2 If the length requirements of the journal permits it, I suggest to add a more thorough discussion of the results. This includes for example a brief discussion on flavoenzyme phylogeny. To my understanding, OpaC belongs to the well-studied class A Flavin monooxygenases that typically feature mobile flavins, see e.g., PMID: 24361254 or PMID: 19944667. One of these references (or a similar one) should be cited here. This would also provide additional support for the proposed mechanism, as so far all class A monooxygenases exclusively utilize C4a-(hydro)peroxide species for catalysis (other Flavin monooxygenases may instead use N5-oxygen adducts for catalysis, see PMID: 32066967 and PMID: 24162851). I would also consider to add a phylogenetic tree to the SI that includes other characterized members of class A monooxygenases. A similar approach could be used for the P450 epoxidase OpaB. What are the closest characterized homologues of this enzyme?

Response 1.2 We constructed a phylogenetic tree with OpaC and 30 class A monooxygenases and added following paragraph for comments.

Sequence analysis and biochemical investigation revealed that OpaC belongs to the well-studied class A flavin-dependent monooxygenases.³¹ Phylogenetic analysis of representatives from this group (Supplementary Figure 5) indicates the presence of at least three clades. OpaC is located closely together with AspB in the biosynthesis of asperlicins and PhqK in that of paraherquamides. AspB catalyzes the hydroxylation at C3 of an indole moiety, resulting in the formation of a hexahydropyrrolo[2,3-b]indole framework.³² PhqK converts an indole ring to a 2-keto indoline ring via a postulated arene oxide intermediate.³³ In both cases, C4a-hydroperoxidflavin species was proposed as active oxygen, being in consistent with our hypothesis.

In addition, we also added a phylogenetic tree for OpaB and homologues and gave short comments.

Phylogenetic analysis of OpaB with 51 known P₄₅₀ enzymes from bacteria and fungi indicates clearly the presence of different clades (Supplementary Figure 4), The bacterial P₄₅₀ enzymes catalyzing diverse reactions, including the two hydroxylases P450cin and P450cam with trace oxepin formation activity,²⁹ build a distinct clade from fungal enzymes. The majority of the fungal P₄₅₀s in the phylogenetic tree catalyzes hydroxylations of diverse substrates. OpaB is located near to the epoxidase AtaY, but far away from the oxepinase AtaF, both involved in the biosynthesis of acetylarnotin in *Aspergillus terreus*.³¹

1.3 The oxepin-epoxide tautomerization should be discussed better. Typically, arene oxides are in rapid spontaneous equilibrium with their oxepins. Which compound is favored depends on the solvent and conjugants mainly in alpha position to my understanding (see DOI: 10.1002/anie.196703851, this reference should also be cited). The authors should discuss if it is known to what extent the different oxepinamides are in equilibrium with their epoxide tautomers. Compound stability might be an important clue here, as epoxides are more reactive and unstable than the oxepins. I would expect the 3H-oxepinamides such as oxepinamide F to be more stable, as the exo double bond formation should completely prevent tautomerization to the epoxide in contrast to the 1H-oxepinamides such as oxepinamide H. This might be a similar strategy as employed by PaaG of phenylacetate degradation and tropone natural product biosynthesis that also forms a stable 3H-oxepin moiety from a labile 1H-oxepin (PMID: 31689071).

Response 1.2 Many thanks for your suggestion. We added following sentences on page 5 for discussion.

The oxepin form is expected to be more stable at room temperature than its arene oxide.²⁸ Only the 1*H*-oxepin **4** was detected as OpaB product in this study, differing from the products of the phenylacetate degradation by PaaG and in the tropone biosynthesis. In those cases, a stable 3*H*-oxepin moiety was formed from a labile 1*H*-oxepin.²⁹ On the other hand, both 1*H*- and 3*H*-oxepin derivatives listed in Fig. 1 were isolated as stable fungal metabolites.

1.4 Minor points:

1.4.1• Spelling of oxepine in general should be oxepin? Are both valid?

Response 1.4.1 Both oxepine and oxepin are valid. We used oxepin instead of oxepine in the revised manuscript and also changed “3*H*-oxepin formation” in the title.

1.4.2• Abstract: correct “oxapinamide”.

Response 1.4.2 We have corrected to “oxepinamide” in the abstract.

1.4.3• p. 2 first paragraph: correct to “potential agents for the treatment of..., and inflammation”.

Response 1.4.3 We have corrected to “potential agents for the treatment of..., and inflammation” in first paragraph on page 2.

1.4.4• p. 2 first paragraph: Please write fumiquinazoline in full before using the abbreviation FQF.

Response 1.4.4 We changed to “fumiquinazoline (FQF)” in the first paragraph on page 2.

1.4.5• p. 2 second paragraph: correct to “structural features”.

Response 1.4.5 We have corrected to “structural features” in the second paragraph on page 2.

1.4.6• p. 3, Results: write “*A. ustus*” in italics.

Response 1.4.6 We have written “*A. ustus*” in italics in Results on page 3.

1.4.7• Fig. 1: the compound number for oxepinamide F could be added.

Response 1.4.7 We have added the compound number (**1**) for oxepinamide F in Figure 1.

1.4.8• Fig. 2: Please indicate the wavelength of the shown chromatograms.

Response 1.4.8 We added the wavelength to the legend of Figure 2.

1.4.9• Fig. 3: The conversion of the epoxide to the oxepin should be fully reversible under physiological conditions, which should be correctly depicted here (i.e. an enzyme should normally not be needed). Or is there any clear reason/evidence for why this isomerization is irreversible? Please correct and discuss, see also comments and suggested references above.

Response 1.4.9 we changed as suggested. See also **Response 1.1**.

1.4.10• p. 3, first line: Correct to “...we sequenced the *A. ustus*...”

Response 1.4.10 We have changed the first line on page 3 to "...we sequenced the *A. ustus*..."

1.4.11• p. 3, second last paragraph: "m/z 362.1888". I suggest to delete the last digit and round the number to 362.189, as typically such a high mass accuracy is not achieved (also apply to other m/z values).

Response 1.4.11 We have deleted the last digit of all m/z values in this manuscript and supporting information.

1.4.12• p. 4, last sentence: This description of the catalytic mechanism of OpaB does not fit the mechanism shown in Fig. 3c. To my understanding, epoxidation could either be catalyzed by an electrophilic hydroxyperoxo-ferric iron (Fe^{III}-OOH) or by an electrophilic oxoferryliron (Fe^{IV}=O), which should be pointed out here and also in the figure legend (see for example PMID: 9520404 or PMID: 15365901).

Response 1.4.12 We improved Fig. 3c, rephrased this paragraph as following and added Refs PMID: 15365901, 196703851 and one additional Publication for citation.

In analogy to other P₄₅₀ catalyzed alkane hydroxylations and alkene epoxidations,^{26, 27} we postulated the electrophilic oxoferryliron (Fe^{IV}=O) as the active oxygen intermediate in the OpaB reaction. Attacking of the oxoferryliron species by the nucleophilic benzene double bond in **3** would result in the formation of an arene oxide, which is in rapid spontaneous equilibrium with the oxepin **4** through 6 π electrocyclic ring opening (Fig. 3c).^{16, 28}

1.4.13• Fig. 4: add wavelength to HPLC chromatograms. Also, the enzyme does not appear to follow typical Michaelis-Menten kinetics, there seems to be some sort of inhibitory effect with substrate conditions above 0.5 mM. This should at least be pointed out. Please also use the term "apparent KM" here.

Response 1.4.13 We have added wavelength to legends in Figures 3 and 4 and also pointed out the inhibitory effect with substrate conditions above 0.5 mM as well as used the term "apparent KM" in this Figure as mentioned below:

"OpaC reaction with **4** in the presence of NADPH does not follow typical Michaelis-Menten kinetics and inhibition was observed with substrate more than 0.5 mM. An apparent *KM* value at 0.15 \pm 0.006 mM and a turnover number (*k_{cat}*) at 0.25 \pm 0.003 s⁻¹ (Fig. 4c) were calculated by using data obtained with **4** of up to 0.5 mM."

1.4.14• p. 5, first paragraph: correct to monooxygenase. Also, what does "planer" mean? Please change to "...5 shares the same UV visible light absorption and mass spectral features with 2".

Response 1.4.14 We have corrected to "monooxygenase" in the first paragraph on page 5 and changed the sentence to "...5 shares the same UV visible light absorption and mass spectral features with 2" in the first paragraph too. We also corrected the "planer structures" to "planar structures" on page 4.

1.4.15• p. 5 second paragraph: correct to "epimerization".

Response 1.4.15 We have corrected it to "epimerization" in the second paragraph on page 5.

1.4.16• p. 6: correct to "phenylacetic acid". Also, this statement is somewhat unclear, I suggest changing to "Oxepin rings also play important roles in the bacterial degradation of phenylacetic acid and the biosynthesis of tropone natural products that both depend on the

multicomponent epoxidase PaaABCE and oxepin isomerase PaaG” (consider adding PMID: 31689071 in addition to ref 29).

Response 1.4.16 Thanks a lot for your comments. We have changed the description to “Oxepin rings also play important roles in the bacterial degradation of phenylacetic acid and the biosynthesis of tropone natural products that both depend on the multicomponent epoxidase PaaABCE and oxepin isomerase PaaG” and also added the reference PMID: 31689071.

1.4.17• p. 6: The following statement should be removed or revised “Thus, OpaB represents the first single P450 to catalyze the direct conversion of a benzene to an oxepine ring by one step conversion.” Basically every enzyme that epoxidizes an aromatic ring also generates an oxepin ring that is spontaneously formed from the epoxide, see also comments above.

Response 1.4.17 We agree and deleted the statement “Thus, OpaB represents the first single P450 to catalyze the direct conversion of a benzene to an oxepin ring by one step conversion.”

1.4.18• p. 6: correct to “...isomerase from microorganisms involved...”

Response 1.4.18 We have corrected to “...isomerase from microorganisms involved...” on page 6.

1.4.19• Fig. 5: add wavelength to HPLC chromatograms.

Response 1.4.19 We added wavelength to the legend in Figure 5.

1.4.20• Fig. 6 is not correctly depicted so I cannot assess the content. Please fix.

Response 1.4.20 We tried to have a better position of Figure 6 in the manuscript.

1.4.21• p. 7: “the cultures were extracted with EtOAc, dissolved in DMSO and subjected to HPLC ...” I suspect the authors mean that EtOAc extracts were rotavaped and the extracted compounds then redissolved in DMSO before analysis? Please explain more clearly.

Response 1.4.21 Sorry for the unclear statement. Yes, we mean the extracts. We rephrased now this sentence as following:

“After cultivation in rice medium at 25°C for 7 days, the cultures were extracted with EtOAc, The EtOAc extracts were evaporated, dissolved in DMSO and subjected to HPLC and HPLC-MS for analysis.”

1.4.22• p. 7: Remove space character between 15-*epi*-oxepinamide.

Response 1.4.22 We have removed the space character between 15-*epi*-oxepinamide.

1.4.23• p. 8: correct to “...the CD spectra are shown...”

Response 1.4.23 We have corrected to “...the CD spectra are shown...” on page 8.

1.4.24• SI Table 7: The icons for COSY, NOESY and HMBC couplings are not properly displayed, please fix.

Response 1.4.24 we improved it in SI Table 7

Reviewer #2 (Remarks to the Author):

The manuscript describes the identification of the oxepinamide F biosynthetic gene cluster and characterization of several of the steps in the pathway. The authors used a combination of in vivo and in vitro data to functionally assign the enzymes. For the former, gene inactivation often led to the isolation of biosynthetic intermediates that were thoroughly characterized by spectroscopy. The latter involved the functional assignment of Flavin dependent oxygenase and an unexpected epimerase.

Overall, the scientific rigor is excellent.

One of the key assignments is the P450 protein that catalyzes the insertion of the oxygen atom to make the oxepine that is characteristic of the family. This result along with the other data make this a very interesting paper with high significance and will have a broad impact in the field. The manuscript is, in general, well written. Once the following minor concerns are addressed, the paper is ready for publication.

Many thanks for your positive comments.

2.1 One of the figures was cut off apparently due to formatting? There are other formatting errors that need to be corrected.

Response 2.1 We improved them, see please **Response 1.4.20.** to Reviewer 1.

2.1 The method used for the deuterium wash-in experiment should be included in the materials and methods section.

Response 2.2 Thank you very much for your suggestion. We have added following description to Methods:

For deuterium labelling experiment with OpaE, the stock solutions of the assay components in H₂O were diluted with D₂O to a final D₂O/H₂O ratio of 9:1 in 50 mM Tris-HCl. The reaction mixtures were incubated and analyzed on LC-MS as described for standard assays.

Reviewer #3 (Remarks to the Author):

This is a very nice and well-written manuscript describing the biosynthesis of a class of natural products from *Aspergillus*. However, it is my opinion that this work is better suited for a more specialised journal such as *Antimicrobial Agents and Chemotherapy*. The approaches used are well-known, and the biosynthesis of similar compounds have been elucidated. There are no unusual enzymatic reactions reported here. Thus while this work is rigorous and well done, I do not believe that it is of interest to the very general readership of *Nature Communications*.

Response 3

As mentioned by Reviewer 1, we believe that our work will be of high interest to a broader scientific community, e.g., researchers interested in natural product biosynthesis, medicinal chemistry or enzymology.

Reviewer #1 (Remarks to the Author):

The revised manuscript by Zheng et al. has been significantly improved and all my concerns were properly addressed. I thus recommend publication of the manuscript. Before publication, a few typos and syntax errors should be corrected, as pointed out below. In addition, I have a few suggestions.

Minor points:

- Title: Consider changing to "Oxepinamide F biosynthesis involves enzymatic..."
- Abstract: replace pyrimidinone "feature" with "moiety"
- P. 2, 1st paragraph: better write "...with an anthranilyl (Ant) residue in common." Also remove space character for "(FQF)"
- P. 2, 2nd paragraph: correct to "...oxepinamide F and G feature a..."
- p. 3, 3rd paragraph: rephrase, e.g., to "...but not 2 production, indicating that OpaF acts as a methyltransferase..."
- P. 4, 1st paragraph: rephrase to "...4 was detected as OpaB product in this study, which differs from an oxepin intermediate important for both phenylacetate degradation and tropone biosynthesis. In those cases, the isomerase PaaG forms a stable 3H-oxepin from a labile 1H-oxepin".
- In the same paragraph, P450s are often misspelled with 450 in the subscript. Also, correct "enzym" to "enzymes"
- P.5, 2nd paragraph: correct to "C4a-hydroperoxyflavin". Also change sentence, e.g., to "In both cases, the C4a-hydroperoxyflavin species was proposed to serve as oxygen transferring agent, consistent with other class A flavin monooxygenases and our mechanistic proposal."
- P.5, 3rd paragraph: correct to "trace amounts of 5 were converted..."
- Conclusion/Discussion: correct "ohenylacetic acid"
- I also suggest to change the following: "...To the best of our knowledge, this is the first report of an oxepinamide biosynthetic gene cluster that includes the characterization of involved biosynthetic enzymes and reactions steps."
- Figure 6: no electron donor is shown for the OpaB reaction, this should be fixed. Moreover, the cyclization is hard to envisage, also because the orientation of the molecule is flipped in the process. Maybe add some arrows to indicate how cyclization occurs here and/or show an additional intermediate.

Reviewer #2 (Remarks to the Author):

The manuscript is a revised version wherein the authors adequately responded to my minor concerns and improved the quality of the manuscript by carefully considering all of the reviewers comments. A concern that was not spotted in the first review is with respect to the single substrate kinetic analysis of OpaC and OpaE and shown in Figures 4 and 5. The data, especially for OpaC shown in Figure 4c, do not appear to fit the Michaelis-Menten (MM) velocity equation very well. Based on the appearance of the plots, it does not appear the authors fit the data to a velocity equation with substrate inhibition despite saying this in the text. OpaE data might also fit the velocity equation with substrate inhibition better than the MM equation, although it is difficult to predict in this case by visual inspection. It is suggested that the authors state whether the data were fitted to the Michaelis-Menten equation or some other velocity equation.

Perhaps more concerning with the kinetic analysis is the extremely low error that is reported in the kinetic parameters, considering the data don't fit the equation very well to begin with. For example, how can your K_m and k_{cat} be so accurate (low error) for OpaC when nearly all the data are outside the fitted curve?

In the experimental, the calculated molar concentrations of enzyme should be provided next to

gram value.

Finally, K_m and k_{cat} are incorrectly written in the text. Both "m" and "cat" should be subscripts

Reviewer #3 (Remarks to the Author):

My only comment was that I thought this was more appropriate for a specialised journal. I was not sure how the enzymology or the biology raised this to the level of Nature Communications. However, this is a decision for the editor.

Reviewer(s)' Comments to Author:

Reviewer #1 (Remarks to the Author):

The revised manuscript by Zheng et al. has been significantly improved and all my concerns were properly addressed. I thus recommend publication of the manuscript. Before publication, a few typos and syntax errors should be corrected, as pointed out below. In addition, I have a few suggestions.

Many thanks for the positive comments and the suggestions.

1. Minor points:

1.1 Title: Consider changing to "Oxepinamide F biosynthesis involves enzymatic..."

Response 1.1. We have changed the title to "Oxepinamide F biosynthesis involves enzymatic D-aminoacyl epimerization, 3H-oxepin formation, and hydroxylation induced double bond migration".

1.2 Abstract: replace pyrimidinone "feature" with "moiety"

Response 1.2 We have replaced it.

1.3 P. 2, 1st paragraph: better write "...with an anthranilyl (Ant) residue in common." Also remove space character for "(FQF)"

Response 1.3 We added the indefinite article "an" before "anthranilyl" and removed the space before "FQF".

1.4 P. 2, 2nd paragraph: correct to "...oxepinamide F and G feature a..."

Response 1.4 We have corrected it.

1.5 p. 3, 3rd paragraph: rephrase, e.g., to "...but not 2 production, indicating that OpaF acts as a methyltransferase..."

Response 1.5 We have rephrased the description to "...but not 2 production, indicating that OpaF acts as a methyltransferase..."

1.6 P. 4, 1st paragraph: rephrase to "...4 was detected as OpaB product in this study, which differs from an oxepin intermediate important for both phenylacetate degradation and tropone biosynthesis. In those cases, the isomerase PaaG forms a stable 3H-oxepin from a labile 1H-oxepin".

Response 1.6 We have rephrased it.

1.7 In the same paragraph, P450s are often misspelled with 450 in the subscript. Also, correct "enzym" to "enzymes"

Response 1.7 We have reunified the writing of P450 and also corrected to "enzymes" in first paragraph on page 4.

1.8 P.5, 2nd paragraph: correct to "C4a-hydroperoxyflavin". Also change sentence, e.g., to "In both cases, the C4a-hydroperoxyflavin species was proposed to serve as oxygen transferring agent, consistent with other class A flavin monooxygenases and our mechanistic proposal."

Response 1.8 We have corrected to “C4a-hydroperoxyflavin” and also changed this sentence in second paragraph on page 5 as your suggestion.

1.9 P.5, 3rd paragraph: correct to “trace amounts of 5 were converted...”

Response 1.9 We have corrected it.

1.10 Conclusion/Discussion: correct “ohenylacetic acid”

Response 1.10 We have corrected it to “phenylacetic acid”.

1.11 I also suggest to change the following: “...To the best of our knowledge, this is the first report of an oxepinamide biosynthetic gene cluster that includes the characterization of involved biosynthetic enzymes and reactions steps.”

Response 1.11 We have changed this sentence based on your suggestion.

1.12 Figure 6: no electron donor is shown for the OpaB reaction, this should be fixed. Moreover, the cyclization is hard to envisage, also because the orientation of the molecule is flipped in the process. Maybe add some arrows to indicate how cyclization occurs here and/or show an additional intermediate.

Response 1.12 Thanks a lot for the valuable advice. We added the electron donor NADPH for OpaB reaction. For better understanding, we changed the orientation of the intermediate to be cyclized and arrows for electron flow of the cyclization in Figure 6 as well.

Reviewer #2 (Remarks to the Author):

2.1 The manuscript is a revised version wherein the authors adequately responded to my minor concerns and improved the quality of the manuscript by carefully considering all of the reviewers comments. A concern that was not spotted in the first review is with respect to the single substrate kinetic analysis of OpaC and OpaE and shown in Figures 4 and 5. The data, especially for OpaC shown in Figure 4c, do not appear to fit the Michaelis-Menten (MM) velocity equation very well. Based on the appearance of the plots, it does not appear the authors fit the data to a velocity equation with substrate inhibition despite saying this in the text. OpaE data might also fit the velocity equation with substrate inhibition better than the MM equation, although it is difficult to predict in this case by visual inspection. It is suggested that the authors state whether the data were fitted to the Michaelis-Menten equation or some other velocity equation.

Perhaps more concerning with the kinetic analysis is the extremely low error that is reported in the kinetic parameters, considering the data don't fit the equation very well to begin with. For example, how can your K_m and k_{cat} be so accurate (low error) for OpaC when nearly all the data are outside the fitted curve?

Response 2.1. Thanks for the positive overall comments and the insightful comments on the kinetic analysis.

Based on your comments, we carried out 6 independent enzyme assays for OpaC and OpaE with substrate concentration from 0.02 mM to 5 mM. Instead of terminating the enzyme assays with MeOH used before, the incubation mixtures were extracted with EtOAc. After evaporation, the residues were dissolved in DMSO and subjected to HPLC for analysis. We added this information to Methods section on page 7.

In these experiments we also got data with very low error, especially in low concentration region.

Kinetic parameters were determined by nonlinear regression using the software GraphPad Prism 6.0. The data for OpaC reaction are more fitted to the typical substrate inhibition velocity equation as showed in Fig. 4c. An apparent K_M value at 0.43 ± 0.04 mM, a turnover number (k_{cat}) at 0.16 ± 0.01 s⁻¹, k_{cat}/K_M at 0.37 mM⁻¹*S⁻¹ and substrate inhibition constant K_i at 0.39 ± 0.03 (Fig. 4c) were calculated by using data obtained with 4 of up to 5 mM. We used standard error of mean (SEM) for data accurate and mentioned it in the Methods section on page 7

For OpaE reaction, the data are more fitted to the Michaelis-Menten equation as showed in Fig. 5c. A K_M value of 1.41 ± 0.05 mM, a turnover number (k_{cat}) of 0.28 ± 0.01 s⁻¹ and k_{cat}/K_M at 0.20 mM⁻¹*S⁻¹ were obtained.

We have updated all the results in Fig. 4 and Fig. 5.

2.2 In the experimental, the calculated molar concentrations of enzyme should be provided next to gram value.

Response 2.2. We added the molar concentrations next to gram value for enzyme assays. The obtained protein amounts from bacterial cultures are still only in gram. We think this make more sense.

2.3 Finally, Km and kcat are incorrectly written in the text. Both "m" and "cat" should be subscripts

Response 2.3. We have changed them to subscript form K_M and k_{cat} as showed in the text.

Reviewer #3 (Remarks to the Author):

My only comment was that I thought this was more appropriate for a specialised journal. I was not sure how the enzymology or the biology raised this to the level of Nature Communications. However, this is a decision for the editor.

REVIEWERS' COMMENTS:

Reviewer #2 (Remarks to the Author):

The kinetic data with the appropriate fitted velocity equation looks much better in the figures. I have no further concerns prior to publication.